# Increased Oxidative Stress and Decreased Citrulline in Blood Associated with Severe Novel Coronavirus Pneumonia in Adult Patients

**DOI:** 10.3390/ijms25158370

**Published:** 2024-07-31

**Authors:** Mitsuru Tsuge, Eiki Ichihara, Kou Hasegawa, Kenichiro Kudo, Yasushi Tanimoto, Kazuhiro Nouso, Naohiro Oda, Sho Mitsumune, Goro Kimura, Haruto Yamada, Ichiro Takata, Toshiharu Mitsuhashi, Akihiko Taniguchi, Kohei Tsukahara, Toshiyuki Aokage, Hideharu Hagiya, Shinichi Toyooka, Hirokazu Tsukahara, Yoshinobu Maeda

**Affiliations:** 1Department of Pediatrics, Graduate School of Medicine, Dentistry and Pharmaceutical Sciences, Okayama University, Okayama 700-8558, Japan; tsukah-h@cc.okayama-u.ac.jp; 2Department of Allergy and Respiratory Medicine, Okayama University Hospital, Okayama 700-8558, Japan; ichiha-e@md.okayama-u.ac.jp (E.I.); atgcuacg@gmail.com (A.T.); 3Department of General Medicine, Graduate School of Medicine, Dentistry and Pharmaceutical Sciences, Okayama University, Okayama 700-8558, Japan; khasegawa@okayama-u.ac.jp (K.H.); hagiya@okayama-u.ac.jp (H.H.); 4Department of Respiratory Medicine, National Hospital Organization Minami-Okayama Medical Center, Okayama 701-1192, Japan; kudoken19800411@yahoo.co.jp (K.K.); mitsumune.sho.mr@mail.hosp.go.jp (S.M.); 5Department of Allergy and Respiratory Medicine, National Hospital Organization Minami-Okayama Medical Center, Okayama 701-0304, Japan; tanimoto.yasushi.tq@mail.hosp.go.jp (Y.T.); kimura.goro.mb@mail.hosp.go.jp (G.K.); 6Department of Gastroenterology, Okayama City Hospital, Okayama 700-0962, Japan; kazunouso@gmail.com; 7Department of Internal Medicine, Fukuyama City Hospital, Fukuyama 721-0971, Japan; dancingqueen121212@gmail.com (N.O.); takata.ichiro@gmail.com (I.T.); 8Department of Infectious Disease, Okayama City Hospital, Okayama 700-0962, Japan; hyamada427@gmail.com; 9Center for Innovative Clinical Medicine, Okayama University Hospital, Okayama 700-8558, Japan; mitsuh-t@cc.okayama-u.ac.jp; 10Department of Emergency, Critical Care and Disaster Medicine, Graduate School of Medicine, Dentistry and Pharmaceutical Sciences, Okayama University, Okayama 700-8558, Japan; pzsm7wbp@cc.okayama-u.ac.jp (K.T.); t_aokage@okayama-u.ac.jp (T.A.); 11Department of General Thoracic Surgery and Breast and Endocrine Surgery, Graduate School of Medicine, Dentistry and Pharmaceutical Sciences, Okayama University, Okayama 700-8558, Japan; toyooka@md.okayama-u.ac.jp; 12Department of Hematology, Oncology and Respiratory Medicine, Graduate School of Medicine, Dentistry and Pharmaceutical Sciences, Okayama University, Okayama 700-8558, Japan; yosmaeda@md.okayama-u.ac.jp

**Keywords:** novel coronavirus disease 2019, pneumonia, hydroperoxide, nitric oxide, reactive oxygen species, citrulline, arginine, asymmetric dimethylarginine

## Abstract

This study investigated the correlation between oxidative stress and blood amino acids associated with nitric oxide metabolism in adult patients with coronavirus disease (COVID-19) pneumonia. Clinical data and serum samples were prospectively collected from 100 adult patients hospitalized for COVID-19 between July 2020 and August 2021. Patients with COVID-19 were categorized into three groups for analysis based on lung infiltrates, oxygen inhalation upon admission, and the initiation of oxygen therapy after admission. Blood data, oxidative stress-related biomarkers, and serum amino acid levels upon admission were compared in these groups. Patients with lung infiltrations requiring oxygen therapy upon admission or starting oxygen post-admission exhibited higher serum levels of hydroperoxides and lower levels of citrulline compared to the control group. No remarkable differences were observed in nitrite/nitrate, asymmetric dimethylarginine, and arginine levels. Serum citrulline levels correlated significantly with serum lactate dehydrogenase and C-reactive protein levels. A significant negative correlation was found between serum levels of citrulline and hydroperoxides. Levels of hydroperoxides decreased, and citrulline levels increased during the recovery period compared to admission. Patients with COVID-19 with extensive pneumonia or poor oxygenation showed increased oxidative stress and reduced citrulline levels in the blood compared to those with fewer pulmonary complications. These findings suggest that combined oxidative stress and abnormal citrulline metabolism may play a role in the pathogenesis of COVID-19 pneumonia.

## 1. Introduction

The novel coronavirus disease 2019 (COVID-19), caused by severe acute respiratory syndrome coronavirus 2 (SARS-CoV-2), was first reported in Wuhan, China, in December 2019, before spreading rapidly worldwide [1,2]. By March 2023, over 760 million people had been infected, with over 6.8 million fatalities worldwide, presenting a significant global public health challenge. The clinical course of COVID-19 varies, ranging from mild upper respiratory symptoms to life-threatening respiratory complications [3]. While vaccination campaigns and antiviral medications have notably reduced the incidence and severity of COVID-19, the emergence of highly contagious mutant strains poses ongoing risks of widespread transmission [4].

The severity of COVID-19 is caused by an excessive release of inflammatory cytokines such as interleukin-6 (IL-6) due to dysregulated innate and adaptive immune systems in the host [5,6]. This hyperactive immune response triggers the activation of macrophages and neutrophils, as well as vascular endothelial dysfunction [7,8]. Vascular endothelial dysfunction plays a role in exacerbating COVID-19, precipitating organ ischemia and circulatory collapse through capillary leakage, thrombus formation, and vasoconstriction [9]. 

Oxidative stress can impair vascular endothelial function, leading to reduced production of endothelial nitric oxide (NO), which promotes vasodilation. In patients with COVID-19, the generation of reactive oxygen species (ROS) in the bloodstream can impair vascular endothelial function, consequently reducing NO production due to endothelial damage [10,11]. Low levels of nitrite/nitrate, byproducts of NO metabolism, have been reported in patients with severe COVID-19 cases [12,13,14], which can contribute to vasodilatory dysfunction and thrombosis in these patients.

While previous studies have reported increased oxidative stress and decreased NO levels in the blood of patients with COVID-19, few studies have investigated the correlation between oxidative stress and NO synthesis in COVID-19. This prospective study aimed to investigate the correlation between oxidative stress and blood amino acids associated with NO metabolism in adult patients hospitalized for COVID-19 with pneumonia.

## 2. Results

Among the 100 patients with COVID-19, 60 were male and 40 were female. The patients’ ages ranged from 21 to 88 years, with a median age of 63.5 years. Body mass index (BMI) varied from 15.2 to 40.2, with a median BMI of 23.8. Overall, 49 patients had a history of smoking, including 30 ex-smokers and 19 current smokers. Their medical histories comprised hypertension (n = 35), diabetes (n = 20), cardiovascular disease (n = 8), chronic lung disease (n = 7), active cancer (n = 2), and rheumatoid arthritis (n = 2).

Hospitalization occurred within a range of −1 to 14 days from the onset of COVID-19 symptoms, with a median of 5 days. Upon admission, common symptoms observed were fever (n = 82), cough (n = 26), tachypnea (>21 beats/min) (n = 22), sore throat (n = 12), malaise (n = 12), dysgeusia (n = 5), sense of smell disability (n = 4), arthralgia (n = 3), dyspnea (n = 3), diarrhea (n = 3), chills (n = 2), sputum production (n = 2), headache (n = 2), nausea (n = 1), low back pain (n = 1), back pain (n = 1), anorexia (n = 1), chest pain (n = 1), and nasal discharge (n = 1).

Systolic blood pressure on admission ranged from 88 to 181 mmHg (median: 126.5 mmHg), while percutaneous oxygen saturation (SpO_2_) ranged from 89 to 100% (median: 96%). Notably, 14 patients exhibited poor oxygenation (SpO_2_ < 93%) upon admission. Blood tests at admission revealed leukocytosis (>10,000/μL) in 3 cases, leukopenia (<4000/μL) in 22 cases, lymphopenia (<1500/μL) in 81 cases, and elevated lactate dehydrogenase (LDH) levels (>250/μL) in 51 patients. Additionally, elevated serum creatinine (>1.0 mg/dL) was observed in 16 cases, high KL-6 (>500 U/mL) in 7 of 78 patients, high ferritin (>300 ng/mL) in 61 of 83 patients, and high D-dimer (>1 μg/mL) was observed in 29 of 99 patients. Likewise, C-reactive protein (CRP) (>0.14 mg/dL) was elevated in 90 of 99 cases. 

Chest CT scans were performed on admission in 92 of 100 patients, with 29 patients showing infiltration in the lung field and 63 patients showing no infiltration. Notably, 42 patients required oxygen inhalation upon admission, with a nasal cannula being used for 34 patients, an oxygen mask for 6 patients, and high-flow nasal cannula oxygen for 2 patients. Out of the 58 patients who did not require oxygen upon admission, 16 required oxygen after admission. However, 4 patients died during hospitalization, all of whom required oxygen upon admission, while the remaining 96 patients recovered.

The patients were categorized into three groups for analysis. Firstly, 63 patients exhibited no lung infiltration findings on chest CT scans upon admission (non-Pneu group), while 29 patients had infiltration findings in the lung (Pneu group). Secondly, 42 patients required oxygen inhalation upon admission (O_2_ group), while 58 patients did not require oxygen inhalation upon admission (non-O_2_ group). Lastly, 42 patients did not require oxygen inhalation throughout their hospitalization (non-Post O_2_ group), whereas 16 patients started oxygen inhalation after hospitalization (Post O_2_ group). Blood data, oxidative stress-related biomarkers, and serum amino acid levels upon admission were compared across these groups.

Upon admission, SpO_2_ levels upon admission were significantly lower in the Pneu group compared to the non-Pneu group. Blood lymphocyte counts were significantly lower in the Pneu group compared to the non-Pneu group. Serum LDH, D-Dimer, CRP, and IL-6 levels upon admission were significantly higher in the Pneu group compared to the non-Pneu group (Table 1). Serum levels of KL-6 and ferritin did not significantly differ between the non-Pneu and Pneu groups.

Serum levels of hydroperoxides were significantly higher in the Pneu compared to the non-Pneu group. Blood citrulline levels were significantly lower in the Pneu group than in the non-Pneu group, and serum nitrite/nitrate, asymmetric dimethylarginine (ADMA), arginine, ornithine, and blood ammonia levels did not differ significantly between the non-Pneu and Pneu groups.

Upon admission, SpO_2_ levels were significantly higher in the O_2_ group compared to the non-O_2_ group (Table 2). The lymphocyte counts were significantly lower in the O_2_ group compared to the non-O_2_ group. Serum LDH, D-dimer, ferritin, CRP, and IL-6 levels were significantly higher in the O_2_ group compared to the non-O_2_ group. Serum levels of KL-6 did not significantly differ between the non-O_2_ and O_2_ groups.

Serum levels of hydroperoxides were significantly higher in the O_2_ group compared to the non-O_2_ group, while serum ADMA levels were significantly lower in the O_2_ group than in the non-O_2_ group. Serum citrulline levels were significantly lower in the O_2_ group compared to the non-O_2_ group, with higher ammonia levels observed in the O_2_ group. Serum levels of nitrite/nitrate, arginine, and ornithine did not significantly differ between the two groups.

Serum LDH, ferritin, and CRP levels were significantly higher in the Post O_2_ group compared to the non-Post O_2_ group (Table 3). Lymphocyte counts, serum KL-6 levels, and D-dimer levels did not exhibit significant differences between the two groups. Serum levels of hydroperoxides were significantly higher in the Post O_2_ group compared to the non-Post O_2_ group, while serum citrulline levels were significantly lower in the Post O_2_ group than in the non-Post O_2_ group. Serum levels of IL-6, nitrite/nitrate, ADMA, arginine, ornithine, and ammonia did not significantly differ between the two groups.

Furthermore, patients with low serum citrulline at admission (<18.0 nmol/mL) (low-Cit group) were analyzed, including those with normal serum citrulline levels at admission (≥18.0 nmol/mL) (non-low-Cit group) (Table 4). A significant difference was observed between the two groups, with 55.6% of patients in the low-Cit group and 21.2% in the non-low-Cit group exhibiting lung infiltration at admission. In terms of oxygen requirement upon admission, 51.9% of patients in the low-Cit group and 38.5% in the non-low-Cit group required oxygen, with no statistically significant difference observed between the groups. Within the low-Cit group, 27.8% of patients did not require oxygen on admission but needed oxygen therapy after admission, compared to 18.8% in the non-low-Cit group, with no significant difference noted between the two groups.

SpO_2_ levels upon admission were lower in the low-Cit group compared to the non-low-Cit group (Table 5). Serum levels of LDH, CRP, and IL-6 upon admission were remarkably higher in the low-Cit group compared to the non-low-Cit group. However, no significant differences were observed in lymphocyte counts, serum KL-6, ferritin, and D-dimer levels upon admission between the two groups. Serum levels of hydroperoxides upon admission were higher in the low-Cit group compared to the non-low-Cit group. Serum arginine levels were significantly lower in the Post O_2_ group compared to the non-Post O_2_ group, but no patients showed hypoargininemia (<31.8 nmol/mL). There were no significant differences in the serum levels of nitrite/nitrate, ADMA, ornithine, and ammonia between the two groups.

Additionally, correlations between serum citrulline levels and serum LDH, CRP, and hydroperoxides upon admission were analyzed (Figure 1). Serum citrulline levels exhibited significant negative correlations with LDH, CRP, and hydroperoxides.

Furthermore, changes in the serum levels of hydroperoxides and citrulline between admission and convalescence were analyzed for each patient. In the Pneu group, serum levels of hydroperoxides significantly decreased upon convalescence compared to admission, whereas serum citrulline levels significantly increased. However, these changes were not observed in the non-Pneu group (Figure 2 and Figure 3). Similarly, in the O_2_ group, serum levels of hydroperoxides significantly decreased upon convalescence compared to admission, with a significant increase in serum citrulline levels; however, these changes were not observed in the non-O_2_ group. Serum levels of hydroperoxides and citrulline in the Post O_2_ group did not exhibit significant changes between admission and convalescence, similar to the findings in the non-Post O_2_ group.

## 3. Discussion

The present prospective cohort study revealed that adult patients with COVID-19 experiencing pneumonia or poor oxygenation exhibited higher levels of oxidative stress and lower levels of citrulline in their blood upon admission, which showed significant correlations with markers of COVID-19 severity, such as serum LDH and CRP levels. Additionally, the significant negative correlation observed between citrulline and hydroperoxides in the blood suggests that increased oxidative stress and abnormal citrulline metabolism might contribute to the pathogenesis of COVID-19 pneumonia. Developing parameters that can facilitate the early assessment and stratification of the risk of severe pneumonia in COVID-19 patients is an important challenge. Previous studies have reported the usefulness of the partial pressure of arterial oxygen (PaO_2_)/fraction of inspiratory oxygen (FiO_2_) ratio [15], the SpO_2_/FiO_2_ ratio [16], alveolar–arterial gradient [17], and IL-6 [18] as parameters for predicting severity in COVID-19 patients. 

Prior studies have reported a marked increase in the levels of oxidants/free radicals in the blood of patients with COVID-19 compared to those without the disease [19,20,21]. ROS are unstable in nature and difficult to detect directly. Hydroperoxides are oxygen metabolites of lipids, proteins, amino acids, and nucleic acids, which are biomarkers of blood oxidative stress [22,23]. During the acute phase, levels of hydroperoxide in the blood of patients with COVID-19 were notably higher compared to those without COVID-19 [24,25]. Moreover, patients with COVID-19 admitted to the intensive care unit (ICU) exhibited significantly higher levels of hydroperoxides in the blood than those with non-ICU admissions [25], and these levels decreased as pneumonia improved during the recovery phase [23].

Furthermore, significant correlations were observed between levels of hydroperoxides and inflammatory cytokines in the blood of patients with COVID-19 pneumonia [22,25]. In patients with moderate/severe COVID-19, a significant decrease in blood hydroperoxides was observed during the recovery period compared to the acute phase [21,26]. Conversely, some studies of patients with moderate/severe COVID-19 found no significant correlation between various oxidative stress parameters and COVID-19 severity [27,28]. Despite limited evidence linking the levels of oxidative stress markers to the severity of COVID-19 pneumonia, our study demonstrated a significant increase in hydroperoxides in the blood upon admission in the presence of oxygen demand or pneumonia. 

Metabolic imbalances related to amino acids, particularly changes in arginine/citrulline metabolism, have been implicated in the severity of COVID-19 in metabolomic studies [29,30,31,32,33,34]. In particular, decreased plasma citrulline levels during the acute phase have been consistently observed in patients with COVID-19 across multiple studies, suggesting a role in the susceptibility and severity of the disease [35,36,37,38,39]. Decreased plasma citrulline levels have been reported not only in adult patients with COVID-19 but also in patients with pediatric COVID-19/multisystem inflammatory syndrome (MIS-C) [36]. Furthermore, blood citrulline levels were reported to be significantly associated with COVID-19 severity and serum cytokine levels [40,41,42,43]. 

Citrulline is produced as a byproduct of arginine metabolism within the liver’s urea cycle. Additionally, within the arginine–citrulline pathway in blood vessels and kidneys, Citrulline is synthesized simultaneously with NO by nitric oxide synthase (NOS) using arginine as a substrate. Citrulline is then recycled back to arginine, maintaining NO synthesis in blood vessels. During inflammation, arginine is consumed through the activation of arginase-1 in blood vessels and macrophages. Plausible mechanisms contributing to the reduction in plasma citrulline levels in COVID-19 pneumonia include (i) the depletion of the arginine substrate; (ii) increased arginase-1 activity, which degrades arginine; (iii) the decreased efficiency of the urea cycle in the liver; and (iv) impaired NO production due to reduced NOS activity. 

Studies have reported lower plasma arginine levels and arginine/ornithine ratios, indicative of reduced arginine bioavailability, in adult patients with COVID-19 and patients with pediatric MIS-C compared to those without COVID-19 [36,44]. Arginine deficiency and increased arginase activity have been associated with lymphopenia and lymphocyte dysfunction in patients with COVID-19 experiencing acute respiratory distress syndrome [45,46]. However, in our study, no significant differences in plasma arginine levels, arginine/ornithine ratios, or serum arginase-1 levels were observed upon admission based on the severity of pneumonia or oxygen demand. This suggests that hypocitrullinemia might not solely be attributed to arginine depletion or degradation. 

Furthermore, citrulline production may decrease due to the impaired efficiency of the urea cycle in the liver. In this study, patients requiring oxygen upon admission were found to exhibit increased plasma ammonia levels, which is metabolized within the urea cycle. The presence of underlying chronic liver disease comorbidities may contribute to hypocitrullinemia. Regrettably, serum transaminase levels at admission and comorbidities of chronic liver disease were not included in the questionnaire.

NO plays an important role in vasodilation and is involved in the inflammatory progression of COVID-19 [12,13,14]. NO synthesis is facilitated by NOS, and its activity is regulated by the endogenous NOS inhibitor ADMA. Elevated ADMA levels in the blood have also been associated with severe COVID-19 and in-hospital mortality [44,47,48,49]. Other clinical investigations found no significant difference in blood nitrite levels between patients with COVID-19 and those without [21,25]. However, nitrite/nitrate and ADMA levels in the blood upon admission were not significantly elevated in patients with COVID-19 experiencing pneumonia or oxygen demand in this study. This could be attributed to the mild-to-moderate pneumonia status of the participants in this study, who did not require mechanical ventilation or oxygen therapy upon admission.

Additionally, our study showed a significant negative correlation between levels of citrulline and hydroperoxides in the blood. Nonetheless, the direct causal relationship between oxidative stress and citrulline remains unclear. Increased oxidative stress can lead to citrulline deficiency through increased arginase production in macrophages and vascular endothelial cells and increased ADMA levels in the blood [50,51]. Citrulline might participate in defense mechanisms against oxidative stress associated with severe pneumonia [52,53].

Our study has some limitations. Firstly, patients requiring mechanical ventilation were excluded from the teprenone clinical trials, resulting in a study population with predominantly mild-to-moderate disease severity, with only 4 fatal cases out of 100. Moreover, the sample size was relatively small, necessitating further studies with a larger cohort, including patients with severe disease. Future studies should focus on careful case selection; matching with healthy and disease controls; and consideration of confounding variables, such as patient demographics, comorbidities, and pharmacological interventions, in a sufficiently large cohort of patients. Secondly, variations in each patient’s clinical background, environment, diet, duration of illness from onset to admission, and pharmaceutical interventions before and during the study could act as confounding factors affecting biomarker analysis. In addition, all patients in this study were Japanese adults; therefore, the results may not be generalizable to patients of other races. Thirdly, some patients with false-negative results may have been excluded due to sampling timing in the SARS-CoV2 reverse transcription polymerase chain reaction (RT-PCR) test or the performance of the RT-PCR test kit at individual inpatient facilities.

## 4. Materials and Methods

This study adhered to the principles outlined in the Declaration of Helsinki, and its protocol was approved by the Okayama University Certified Review Board (Approval No.: CRB20-001). Prior to their enrollment in this study, written informed consent was obtained from each patient. 

### 4.1. Patients

This multicenter, observational, prospective cohort study was performed as part of a phase II randomized controlled clinical trial to investigate the efficacy of a combination of teprenone for COVID-19 pneumonia (jRCTs061200002) [54]. We recruited 100 hospitalized adult Japanese patients diagnosed with COVID-19 between July 2020 and August 2021 after obtaining informed consent. Participating inpatient facilities included Okayama University Hospital, National Hospital Organization Okayama Medical Center, Okayama City Hospital, Fukuyama City Hospital, and National Hospital Organization Minami Okayama Medical Center.

The inclusion criteria were as follows: individuals aged ≥20 years, having a confirmed COVID-19 diagnosis, and a fever of 37 °C or higher. Diagnosis of SARS-CoV-2 infection was confirmed through RT-PCR testing of nasopharyngeal swab specimens. Exclusion criteria comprised patients with a recent history of oral teprenone within 2 weeks of obtaining informed consent, those requiring ventilator management or extracorporeal membrane oxygenation at the time of consent, pregnant or breastfeeding patients, and those with a history of severe drug allergy to teprenone.

Participants were randomly assigned in a 1:1 ratio to either a control group (n = 50) or a teprenone-treated group (n = 50). The control group received standard therapy, while the teprenone-treated group received oral teprenone 50 mg three times daily for 10 days in addition to standard therapy. All participants underwent close monitoring from admission until the end of the study. The primary endpoint was time to resolution of fever, while the secondary endpoints included intubation rate, mortality, and safety. Notably, a previous clinical study did not demonstrate significant efficacy of teprenone in terms of the primary and secondary endpoints [54]. Consequently, a sub-analysis was conducted using serum samples obtained during this clinical trial to measure specific biomarkers.

Each patient’s background data at the time of admission, such as gender, age, physical condition, medical history, and smoking status, as well as symptoms and vital signs, including SpO_2_, were recorded. Information on oxygen inhalation and artificial ventilation management during hospitalization, medications, and prognosis was recorded at the time of discharge. Chest CT scans, initiation of oxygen inhalation, and permission for discharge were decided by the attending physician.

Patients were divided into three comparison groups for analysis, as described below. The first comparison group consisted of patients who exhibited no lung infiltration findings on chest CT scans upon admission (non-Pneu group) and patients who had infiltration findings in the lung (Pneu group). Patients without CT scans were excluded. The second comparison group consisted of patients who required oxygen inhalation upon admission (O_2_ group) and patients who did not require oxygen inhalation upon admission (non-O_2_ group). The O_2_ group included patients who received oxygen via a nasal cannula, oxygen mask, and high-flow nasal cannula. This study did not include patients who were mechanically ventilated on admission. The third comparison group consisted of patients who did not require oxygen inhalation throughout their hospitalization (non-Post O_2_ group) and patients who started oxygen inhalation after hospitalization (Post O_2_ group). The Post O_2_ group included patients who received oxygen via a nasal cannula, oxygen mask, or high-flow nasal cannula but did not include patients in whom mechanical ventilation was initiated. 

### 4.2. Collection of Serum Samples and Patient Data

Following informed consent, blood samples for this clinical study were collected during routine blood tests conducted throughout the patient’s hospitalization. Blood samples were drawn into serum separator tubes within the first 2 days of admission (days 0–2) and approximately 10 days post-admission at each participating hospital. All samples were anonymized. The serum obtained post-centrifugation was aliquoted into cryogenic tubes and frozen at −80 °C at each participating institution until further analysis. Subsequently, the frozen serum samples, along with relevant patient history and blood test data, were transported to Okayama University Hospital. 

### 4.3. Measurement of Each Biomarker

Lymphocyte counts, LDH, KL-6, ferritin, D-dimer, and CRP levels were measured using automated analyzers at each participating hospital.

IL-6, ADMA, and arginase-1 were quantified using enzyme-linked immunosorbent assay kits, following the manufacturer’s instructions as reported before [55,56,57,58] (IL-6: R&D Systems, Inc., Minneapolis, MN, USA; ADMA: Immundiagnostik AG, Manchester, NH, USA; arginase-1: RayBiotech, Inc., Peachtree Corners, GA, USA). Serum level of hydroperoxides was measured by the diacron-reactive oxygen metabolite (d-ROM) kit using the Free Radical Analytical System (FREE Carrio Duo, Wismerll Co., Ltd., Tokyo, Japan) [59,60,61] that applies Fenton and Haber–Weiss reactions, where one U.CARR (unit of hydroperoxides value) is equivalent to 0.08 mg/dL of H_2_O_2_. Nitrite/nitrate levels were quantitatively determined using the Griess reaction, according to the manufacturer’s instructions (R&D Systems, Inc., Minneapolis, MN, USA). Serum arginine, citrulline, ornithine, and ammonia levels were quantitatively measured using high-performance liquid chromatography at a commercial laboratory (Biomedical Laboratories, Tokyo, Japan). Due to the insufficient volume of samples in some patients, we were able to measure the above amino acids in the serum of 79 patients.

### 4.4. Statistical Analysis

All data analyses were performed using GraphPad Prism (version 7.0; GraphPad Software Inc., La Jolla, CA, USA). Comparative analysis for each parameter was performed by excluding cases with missing values and limiting the cases with variable data. Nonparametric continuous data were presented as medians and interquartile ranges for quantitative variables. 

The Mann–Whitney U-test was applied to compare the two groups with non-normally distributed data. The Wilcoxon signed-rank test was used to evaluate pre- and post-treatment biomarker changes within groups. Categorical variables were expressed as frequencies and percentages, and between-group comparisons were performed using the chi-square test or Fisher’s exact test. Pearson’s correlation analysis was performed for correlation assessments. Statistical significance was set at *p* < 0.05, with no adjustments made for multiple comparisons.

## 5. Conclusions

In conclusion, this prospective study revealed the correlation between oxidative stress and blood amino acids associated with NO metabolism in adult patients hospitalized for COVID-19 with pneumonia. Patients with COVID-19 suffering from pneumonia or poor oxygenation exhibited higher blood levels of oxidative stress and lower citrulline levels compared to those without complications, which may contribute to the risk of developing COVID-19 pneumonia. Future studies should focus on careful case selection; matching with healthy and disease controls; and consideration of confounding variables, such as patient demographics, comorbidities, and pharmacological interventions, in a sufficiently large cohort of patients.

## Figures and Tables

**Figure 1 ijms-25-08370-f001:**
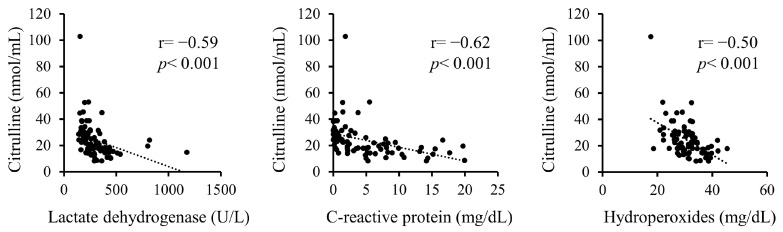
Correlations between serum citrulline levels and serum levels of LDH, CRP, and hydroperoxides on admission.

**Figure 2 ijms-25-08370-f002:**
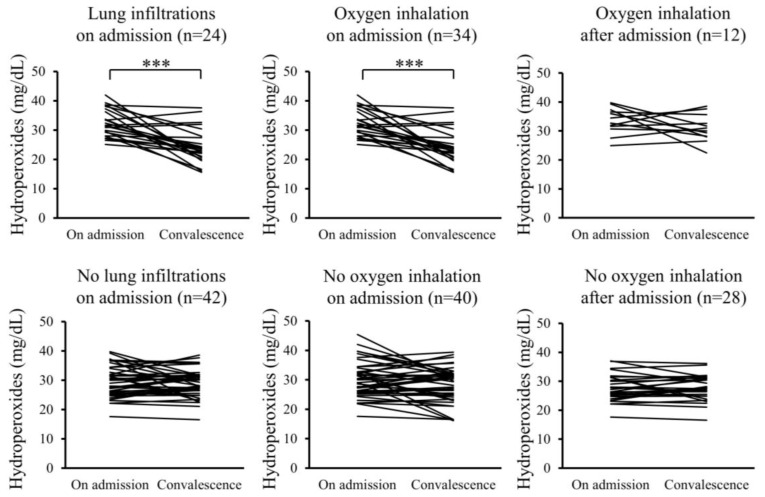
Changes in serum levels of hydroperoxides between admission and convalescence in each patient. *** *p* < 0.001.

**Figure 3 ijms-25-08370-f003:**
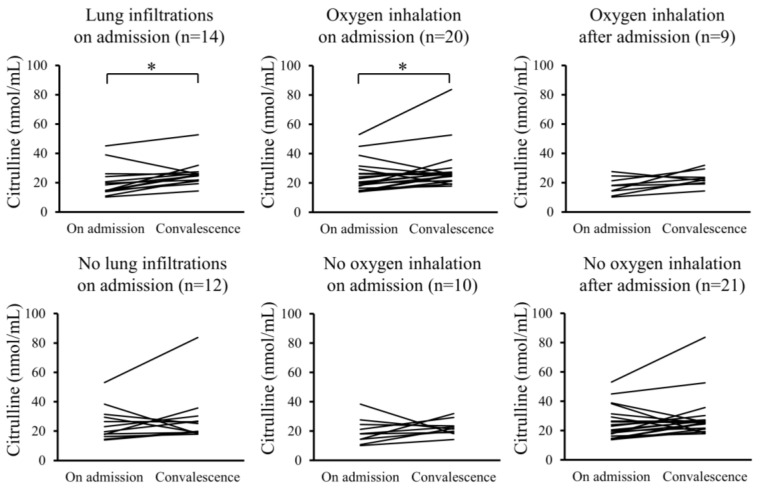
Changes in serum levels of citrulline between admission and convalescence in each patient. * *p* < 0.05.

**Table 1 ijms-25-08370-t001:** Comparison of each biomarker on admission between the patients with and without lung infiltrations on chest CT.

Biomarkers	Lung Infiltration (n = 29)	No Infiltration (n = 63)	*p*-Value
Saturation of percutaneous oxygen (%)	95 (91–96)	97 (95–98)	**<0.001**
Lymphocyte counts (/µL)	925 (530–1184)	1150 (828–1450)	**0.025**
Lactate dehydrogenase (U/L)	362 (293–446)	218 (180–284)	**<0.001**
KL-6 (U/mL)	291 (229–443)	205 (163–268)	0.10
Ferritin (ng/mL)	1049 (717–2501)	607 (258–985)	0.38
D-dimer (μg/mL)	1.2 (0.7–2.0)	0.5 (0.0–1.0)	**<0.001**
C-reactive protein (mg/dL)	7.9 (5.0–14.2)	1.4 (0.3–3.4)	**<0.001**
Interleukin-6 (pg/mL)	62 (26–93)	23 (15–45)	**0.001**
Hydroperoxides (mg/dL)	32 (29–36)	30 (21–33)	**0.031**
Nitrite/nitrate (µmol/L)	28 (16–37)	28 (21–35)	0.47
Asymmetric dimethylarginine (µmol/L)	0.23 (0.04–0.32)	0.26 (0.09–0.44)	0.072
Arginine (nmol/mL)	134 (112–154)	135 (111–145)	0.94
Ornithine (nmol/mL)	83 (75–106)	80 (61–100)	0.68
Citrulline (nmol/mL)	15 (13–20)	25 (18–31)	**0.00** **2**
Ammonia (nmol/mL)	117 (100–145)	118 (104–139)	0.42
Arginase-1 (ng/mL)	152 (101–241)	157 (123–228)	0.38

Significant *p*-values are in bold. Data are expressed as median (interquartile range).

**Table 2 ijms-25-08370-t002:** Comparison of each biomarker on admission between the patients with oxygen inhalation on admission and those without oxygen inhalation on admission.

Biomarkers	Oxygen Inhalation on Admission (n = 42)	No Inhalation on Admission (n = 58)	*p*-Value
Saturation of percutaneous oxygen (%)	95 (93–97)	97 (95–98)	**<0.001**
Lymphocyte counts (/µL)	910 (643–1219)	1151 (835–1450)	**0.037**
Lactate dehydrogenase (U/L)	334 (281–409)	209 (179–249)	**<0.001**
KL-6 (U/mL)	266 (202–374)	210 (173–274)	0.28
Ferritin (ng/mL)	738 (434–1083)	588 (220–65,250)	**0.037**
D-dimer (μg/mL)	1.1 (0.7–1.4)	0.5 (0.0–0.8)	**<0.001**
C-reactive protein (mg/dL)	5.3 (2.1–10.4)	0.7 (0.2–3.3)	**<0.001**
Interleukin-6 (pg/mL)	44 (25–93)	23 (15–40)	**0.004**
Hydroperoxides (mg/dL)	31 (28–36)	30 (25–33)	**0.012**
Nitrite/nitrate (µmol/L)	30 (21–36)	28 (18–40)	0.59
Asymmetric dimethylarginine (µmol/L)	0.10 (0.04–0.28)	0.31 (0.21–0.48)	**<0.001**
Arginine (nmol/mL)	136 (115–149)	131 (108–145)	0.91
Ornithine (nmol/mL)	88 (76–118)	75 (58–96)	0.16
Citrulline (nmol/mL)	19 (15–25)	25 (18–31)	**<0.001**
Ammonia (nmol/mL)	139 (116–170)	110 (95–125)	**0.035**
Arginase-1 (ng/mL)	156 (116–247)	140 (105–228)	0.76

Significant *p*-values are in bold. Data are expressed as median (interquartile range).

**Table 3 ijms-25-08370-t003:** Comparison of each biomarker at admission between patients with oxygen inhalation after admission and those without oxygen inhalation after admission among patients without oxygen inhalation at admission.

Biomarkers	Oxygen Inhalation after Admission (n = 16)	No Oxygen Inhalation after Admission (n = 42)	*p*-Value
Saturation of percutaneous oxygen (%)	96 (95–97)	97 (96–98)	0.063
Lymphocyte counts (/µL)	1105 (843–1426)	1172 (839–1506)	0.78
Lactate dehydrogenase (U/L)	228 (216–304)	204 (171–238)	**0.013**
KL-6 (U/mL)	224 (187–295)	207 (156–267)	0.16
Ferritin (ng/mL)	589 (297–153,000)	481 (179–61,000)	**0.00** **2**
D-dimer (μg/mL)	0.6 (0.3–1.1)	0.0 (0.0–0.7)	0.063
C-reactive protein (mg/dL)	2.0 (1.2–4.1)	0.4 (0.2–2.5)	**0.021**
Interleukin-6 (pg/mL)	32 (19–51)	18 (13–36)	0.16
Hydroperoxides (mg/dL)	32 (29–36)	28 (25–32)	**0.014**
Nitrite/nitrate (µmol/L)	28 (15–42)	27 (19–36)	0.53
Asymmetric dimethylarginine (µmol/L)	0.27 (0.20–0.31)	0.33 (0.22–0.53)	0.18
Arginine (nmol/mL)	116 (106–136)	135 (111–147)	0.056
Ornithine (nmol/mL)	92 (69–102)	69 (56–87)	0.52
Citrulline (nmol/mL)	18 (15–27)	28 (18–31)	**0.031**
Ammonia (nmol/mL)	103 (97–115)	111 (95–129)	0.056
Arginase-1 (ng/mL)	135 (103–256)	157 (105–223)	0.81

Significant *p*-values are in bold. Data are expressed as median (interquartile range).

**Table 4 ijms-25-08370-t004:** Comparison of each biomarker between patients with low serum citrulline levels and patients with normal serum citrulline levels at admission.

Comparative Groups	Low Citrulline (<18.0 nmol/mL)	Normo-Citrulline (≥18.0 nmol/mL)	*p*-Value
Lung infiltrations on admission/all cases	15/27 (55.6%)	11/52 (21.2%)	**0.002**
Oxygen inhalation on admission/all cases	14/27 (51.9%)	21/52 (40.4%)	0.33
Oxygen inhalation after admission/no inhalation on admission	5/13 (38.5%)	6/31 (19.4%)	0.18

Significant *p*-values are in bold. Data are expressed as n (%).

**Table 5 ijms-25-08370-t005:** Comparison of each biomarker at admission between patients with oxygen inhalation after admission and those without oxygen inhalation after admission among patients without oxygen inhalation at admission.

Biomarkers	Low Citrulline (<18.0 nmol/mL) (n = 27)	Normo-Citrulline (≥18.0 nmol/mL) (n = 52)	*p*-Value
Saturation of percutaneous oxygen (%)	95 (94–96)	97 (95–98)	**0.008**
Lymphocyte counts (/µL)	890 (620–1270)	1070 (778–1450)	0.19
Lactate dehydrogenase (U/L)	314 (271–422)	223 (179–307)	**<0.001**
KL-6 (U/mL)	247 (215–317)	202 (163–289)	0.11
Ferritin (ng/mL)	927 (434–5233)	730 (241–1937)	0.41
D-dimer (μg/mL)	0.8 (0.2–1.2)	0.7 (0.0–1.2)	0.45
C-reactive protein (mg/dL)	6.4 (4.2–12.0)	1.2 (0.3–5.1)	**<0.001**
Interleukin-6 (pg/mL)	49 (27–84)	25 (15–52)	**0.011**
Hydroperoxides (mg/dL)	35 (30–37)	29 (26–32)	**<0.001**
Nitrite/nitrate (µmol/L)	21 (18–38)	28 (21–36)	0.61
Asymmetric dimethylarginine (µmol/L)	0.24 (0.09–0.32)	0.27 (0.10–0.48)	0.14
Arginine (nmol/mL)	121 (108–135)	140 (119–152)	**0.00** **4**
Ornithine (nmol/mL)	77 (59–88)	88 (67–101)	0.12
Ammonia (nmol/mL)	115 (100–139)	124 (104–144)	0.40
Arginase-1 (ng/mL)	157 (110–234)	152 (106–231)	0.97

Significant *p*-values are in bold. Data are expressed as median (interquartile range).

## Data Availability

The original contributions presented in the study are included in the article; further inquiries can be directed to the corresponding author/s.

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
