# Peer review of "Increased Oxidative Stress and Decreased Citrulline in Blood Associated with Severe Novel Coronavirus Pneumonia in Adult Patients"

_ijms, 2024, doi:10.3390/ijms25158370_

Round 1
Reviewer 1 Report
Comments and Suggestions for Authors
The authors can expand the chapter with other similar studies that investigate the role of other parameters in COVID-19 patients, such as P/F ratio (DOI: 10.1016/j.ejim.2021.06.002 ), saturation ( DOI: 10.1016/j.ejim.2021.12.015 ), alveolar-arterial gradient ( DOI: 10.1016/j.ejim.2021.12.015 )
alveolar-arterial gradient ( DOI: 10.3390/idr14030050 ) etc.
Results:
Variables were expressed as numerator/denominator (percentage) where possible, median followed by interquartile range 25%-75%.
The p-values (in the text and in the table) are not variables and don't appear as SI. If you wish, you can at least write the p-value as "< 0.001" (and not "p= 4.7×10-9").
Table 2-3: Did you look at other variables such as IL-6, P/F, SpO2, ?
Did you investigate any correlation between the patients who died and the levels of the variables on admission? This type of analysis may be useful for the readers.
Materials and Methods:
Please explain the inclusion/exclusion criteria of all groups ("O2" and "non-O2", "post-O2", "Pneu" etc).
Explain when the variables were collected (at admission, at discharge, etc.).
Average
Author Response
To Reviewer 1:
We are grateful to Reviewer 1 for the critical comments and suggestions that have helped us improve our paper. As indicated in the following responses, we have considered all these comments and suggestions while revising our manuscript.
Point 1: The authors can expand the chapter with other similar studies that investigate the role of other parameters in COVID-19 patients, such as P/F ratio (DOI:10.1016/j.ejim.2021.06.002 ), saturation (DOI:10.1016/j.ejim.2021.12.015), alveolar-arterial gradient (DOI:10.3390/idr14030050 ) etc.
Response 1: Thank you for the important suggestion. As you pointed out, we added the description of other similar studies that investigate the role of other parameters including SpO2 or IL-6 in COVID-19 patients in the Discussion (Line 247-252).
Point 2: Results: Variables were expressed as numerator/denominator (percentage) where possible, median followed by interquartile range 25%-75%. The p-values (in the text and in the table) are not variables and don't appear as SI. If you wish, you can at least write the p-value as "< 0.001" (and not "p= 4.7×10-9").
Response 2: Thank you for your critical comments. We have deleted the p-values ​​in the text and revised the p-values ​​in the tables/figures that were less than 0.001 as "< 0.001." (Table 1-5, Figure 1).
Point 3: Table 2-3: Did you look at other variables such as IL-6, P/F, SpO2?
Response 3: Thank you for your important comments. Unfortunately, the survey for this study did not include the P/F ratio of each patient at admission.
We additionally measured IL-6 concentrations using the remaining serum by ELISA (Line 403-404). As a result, Serum IL-6 levels upon admission were significantly higher in the Pneu group compared to the non-Pneu group (Line 124-125, Table 1). Serum IL-6 levels were significantly higher in the O2 group compared to the non-O2 group (Line 140-141, Table 2). Serum levels of IL-6 upon admission were remarkably higher in the low-Cit group compared to the non-low Cit group (Line 192-193, Table 5). We added descriptions of these results in the text.
We also additionally analyzed SpO2 levels on admission using existing data (Line 99-100). As a result, SpO2 levels upon admission were significantly lower in the Pneu group compared to the non-Pneu group (Line 124-125, Table 1). SpO2 levels were significantly higher in the O2 group compared to the non-O2 group (Line 140-141, Table 2). SpO2 levels upon admission were lower in the low-Cit group compared to the non-low Cit group (Line 192-193, Table 5). We added descriptions of these results in the text.
Point 4: Did you investigate any correlation between the patients who died and the levels of the variables on admission? This type of analysis may be useful for the readers.
Response 4: Thank you for your insightful suggestion. The study included patients with mild to moderate COVID-19 who did not require mechanical ventilation; only four patients died during the study, while the remaining 96 patients recovered. Therefore, we determined that the number of cases was too small to perform statistical analysis.
Point 5: Materials and Methods: Please explain the inclusion/exclusion criteria of all groups ("O2" and "non-O2", "post-O2", "Pneu" etc). Explain when the variables were collected (at admission, at discharge, etc.).
Response 5: Thank you for your critical comments. We added information about when the variables were collected and an explanation of the inclusion/exclusion criteria for the three cohort groups (Line 369-387).
Reviewer 2 Report
Comments and Suggestions for Authors
Thank you for the opportunity to review this outstanding manuscript describing an innovative analysis of oxidative stress and dysregulated amino acid metabolism in adult patients hospitalized with Covid-19 pneumonia. Your study design is sound, and your results presented in a clear and logical fashion, accompanied by helpful tables and graphs. I agree with your discussion points and conclusion. I have only two suggestions:
1. Please clarify sentence 4 of the Abstract ("patients were categorized into two groups...."). In paragraph 5 of the Results, you describe patients being divided into 3 cohorts, based on lung infiltrates, oxygen use on admission, and initiation of oxygen therapy after admission.
2. In the last paragraph of the Discussion (study limitations), please comment on the generalizability of your results, given that all patients were Japanese adults. Are there any major treatment, dietary, or environmental confounders that should be considered?
Author Response
To Reviewer 2:
We are grateful to Reviewer 2 for taking the time to offer us the comments and insights related to the paper. As indicated in the following responses, we have considered all these comments and suggestions while revising our manuscript.
Point 1: Please clarify sentence 4 of the Abstract ("patients were categorized into two groups...."). In paragraph 5 of the Results, you describe patients being divided into 3 cohorts, based on lung infiltrates, oxygen use on admission, and initiation of oxygen therapy after admission.
Response 1: Thank you for the important suggestion. We have revised the description of the categorization of COVID-19 patients into three cohort groups in this study (Lines 37-39).
Point 2: In the last paragraph of the Discussion (study limitations), please comment on the generalizability of your results, given that all patients were Japanese adults. Are there any major treatment, dietary, or environmental confounders that should be considered?
Response 2: Thank you for this insightful suggestion. We added the description that the results may not be generalizable to patients of other races as a limitation (Lines 333-334). As you pointed out, the patient's environment, dietary habits, and treatment for COVID-19 are considered to be important confounding factors in this study. The patient's environment and dietary habits were not included in the questionnaire. The pharmaceutical interventions for COVID-19 before and during the study were not standardized, and medications were selected at the discretion of the attending physician. We have added a description stating that the patient's environment, diet, and treatment were confounding factors in the results of this study (Lines 331-332).
Reviewer 3 Report
Comments and Suggestions for Authors Dear authors, I would like to thank you for the opportunity to review this article, which studies increased oxidative stress and decreased blood citrulline associated with severe coronavirus pneumonia in adult patients. Here are my comments: The INTRODUCTION chapter correctly presents the hypothesis and purpose of the study. RESULTS chapter. There are several situations in which the value of p is written incorrectly. Please fix this. In table number 1, arginase-1 has no unit of measure. The DISCUSSION chapter is well documented, but this study has limitations and questions. The limitations of the study are very well highlighted. The MATERIAL AND METHOD chapter. The number of patients studied is much too small. It is not possible to make a correct statistic. The CONCLUSIONS chapter will highlight more clearly whether the aim of the study has been achieved. I wish you luck!Author Response
To Reviewer 3:
We are grateful to Reviewer 3 for the critical comments and useful suggestions that have helped us improve our paper. As indicated in the following responses, we have considered all these comments and suggestions while revising our manuscript.
Point 1: RESULTS chapter. There are several situations in which the value of p is written incorrectly.
Response 1: Thank you for pointing out the error. We apologize for the incorrect p-values ​​listed in the text. Following suggestions from other reviewers, we have deleted the p-values ​​listed in the text and provided the correct p-values ​​only in the tables (Table 1-5).
Point 2: Please fix this. In table number 1, arginase-1 has no unit of measure.
Response 2: Thank you for your comment. We have added the units of Arginase-1 in Table 1 (Table 1).
Point 3: The MATERIAL AND METHOD chapter. The number of patients studied is much too small. It is not possible to make a correct statistic.
Response 3: Your valuable comment is greatly appreciated. We agree that the small number of patients included in this study is one of its limitations (Lines 325-327). We added a description to the discussion that future cohort studies with larger numbers of patients are needed (Lines 327-330).
Point 4: The CONCLUSIONS chapter will highlight more clearly whether the aim of the study has been achieved.
Response 4: Thank you for your valuable comment. We added the description stating that this prospective study showed that oxidative stress and citrulline are associated with severity in adult patients with COVID-19 (Lines 431-433).
Round 2
Reviewer 1 Report
Comments and Suggestions for Authors
Good job
Reviewer 3 Report
Comments and Suggestions for Authors
Dear authors,
I would like to thank you for the opportunity to review this article, which studies increased oxidative stress and decreased blood citrulline associated with severe coronavirus pneumonia in adult patients.
Thank you for making changes to the article to improve it.
I wish you luck!